# Is Physical Activity an Effective Factor for Modulating Pressure Pain Threshold and Pain Tolerance after Cardiovascular Incidents?

**DOI:** 10.3390/ijerph191811276

**Published:** 2022-09-08

**Authors:** Katarzyna Leźnicka, Maciej Pawlak, Agnieszka Maciejewska-Skrendo, Jacek Buczny, Anna Wojtkowska, Grzegorz Pawlus, Anna Machoy-Mokrzyńska, Aleksandra Jażdżewska

**Affiliations:** 1Department of Physical Education, Academy of Physical Education and Sport, ul. Kazimierza Górskiego 1, 80-336 Gdansk, Poland; 2Department of Physiology and Biochemistry, Poznan University of Physical Education, 61-871 Poznan, Poland; 3Institute of Physical Culture and Health Promotion, University of Szczecin, 70-237 Szczecin, Poland; 4Institute of Psychology, SWPS University of Social Sciences and Humanities, 81-745 Sopot, Poland; 5Department of Experimental and Applied Psychology, Faculty of Behavioural and Movement Sciences, Vrije Universiteit, 1081 HV Amsterdam, The Netherlands; 6Institute of Psychology, SWPS University of Social Sciences and Humanities, 53-238 Wroclaw, Poland; 7Department of Experimental and Clinical Pharmacology, Pomeranian Medical University, 70-111 Szczecin, Poland

**Keywords:** pressure pain threshold, pressure pain tolerance, cardiac patient, pain coping strategy, cardiac rehabilitation, parenting style

## Abstract

The purpose of this study was to investigate whether regular physical activity can alter the pressure pain threshold, pain tolerance, and subjective pain perception in individuals who have experienced a cardiovascular event. The study involved 85 individuals aged 37 to 84 years (*M* = 65.36) who qualified for outpatient cardiac rehabilitation, which consisted of 24 physical training sessions. The patients were all tested twice: on the first and last day of the outpatient cardiac rehabilitation program. Assessments of the pressure pain threshold and pain tolerance were performed with an algometer. To assess the pain coping strategies, the Pain Coping Strategies Questionnaire (CSQ) and parenting styles were measured retrospectively with subjective survey questions. The main results of the study showed that patients achieved significantly higher pressure pain thresholds after a physical training cycle (*p*s < 0.05, η^2^ = 0.05–0.14), but found no differences in the pain tolerance (*p*s > 0.05). A lower preference for the better pain coping strategy explanation (ß = −0.42, *p* = 0.013) and growing up in a family with a less neglectful atmosphere (ß = −0.35, *p* = 0.008) were associated with increased pressure pain threshold after physical training. The results suggest that physical activity is an important factor in modulating the pressure pain threshold.

## 1. Introduction

The intensity, type of pain, and ability to control pain can be influenced by many factors including the location and duration of pain, the patient’s personality, previous pain experiences, life satisfaction, social contacts, and physical activities.

Chest pain is the best-known self-diagnostic factor [1], and it is assumed that the reason a patient sometimes does not feel pain is because they do not recognize it at the level of the peripheral or central nervous system. This hypothesis is supported by the results of previous research, which showed that unrecognized myocardial infarction (MI) was associated with attenuated pain sensitivity [2,3,4]. Some of the findings have also indicated a link between attenuated persons with unrecognized MI and the possibility of identifying such persons by performing a pain sensitivity assessment under experimental conditions, bearing in mind that the larger proportion of MIs are unrecognized in females than in males [5,6,7,8]. A lower tolerance to pain in females than males and a higher percentage of unrecognized MIs in females were observed in a study by Øhrn [9]. The abolition of the feeling of pain is also closely related to the emotional and mental state of a person, embedded in specific interpersonal relations and environmental conditions. In addition, the quality of parenting style represents a potentially important developmental risk factor that may increase the stress reactivity. Family environments characterised by hostility and no interactions of family warmth are among the important risk factors associated with the subsequent occurrence of stress-related health problems [10]. Some researchers have recognized emotional reactivity [11] and biological factors [12] for coping with pain as a potential mechanism for sex differences in pain perception [13]. Coping with pain has a significant impact on clinical outcomes including pain severity and disability [14]. Notably, catastrophising is considered to be the most powerful predictor of pain outcomes, positively correlated with pain intensity and depression, and also with responses to standardised noxious stimuli [15]. The results of the experimental studies suggest that differences in pain sensitivity affect the perception of MI, and have shown that asymptomatic ischemia is associated with attenuated response to pain [8,16]. Furthermore, the body’s vegetative responses are able to modulate the intensity of subjectively perceived pain. It has been reported that patients with lower systolic BP reported higher chest pain scores at symptom onset and during peak acute myocardial infarction (AMI) [17].

In theory and clinical practice, physical activity is considered to be one of the most important elements of pain prevention and treatment. Numerous studies have shown a reduction in the intensity of pain in people who regularly exercise as part of recreational, amateur, and competitive sports [18]. The results of the published scientific research indicate that even low physical activity may modulate the threshold for receiving painful and non-painful stimuli [19].

One of the crucial elements of comprehensive cardiac rehabilitation (CR) is physical activity (PA). Health care organisations recommend that cardiac patients spend 30–60 min performing moderate-intensity physical activity 5 days a week [20,21]. However, only a few patients achieve this level of PA [22]. The health benefits of cardiac rehabilitation are significant and exist regardless of the type of cardiovascular disease [23].

Thus, in this study of patients who participated in a cardiac rehabilitation course, we aimed to evaluate the effects of ordered physical exercises on the pressure pain threshold and pain tolerance as well as subjective pain perception and the possible modulating effects of coping with stress and parenting styles.

## 2. Methods

### 2.1. Study Population

The initial group for this study included 92 patients with a history of myocardial infarction (MI), coronary angioplasty, or cardiac surgery. After examining the immediate threat to life and initial inpatient rehabilitation, patients were referred from the hospital for outpatient cardiological rehabilitation. A cardiologist determined patients who qualified for the second stage of cardiac rehabilitation; however, due to the incomplete data for the tested variables, especially pain tolerance and pain coping strategies, 85 participants, characterised in Table 1, were included in the final sample.

### 2.2. Study Design

Written informed consent was required from all patients to participate in the study. On the first day of rehabilitation, patients reported one hour before treatment, completed the personal questionnaire and the standardised psychological questionnaire, and then the previously described measurements were performed (height, weight, PPT, PTOL, and VAS). On the last day of rehabilitation, one hour after the end of the exercises, the whole procedure was repeated as described in the introduction. All measurements were performed by the same investigator under the same conditions, always in the morning at admission and after the completion of rehabilitation. The second rehabilitation phase lasted a maximum of 12 weeks and included 24 training sessions, each lasting 1 h and consisting of the following.

-interval training on a bicycle ergometer (Aspel CRG 200 v. 312, CardioTEST Alfa System CRG200, AsTER Rehabilitacja Beta System, Szczecin, Poland) with ECG monitoring for 30 min with 3-min load periods and alternation with 2–3-min rest periods;-general exercises (breathing exercises, stretching and relaxation exercises) lasting 30 min.

The participants were informed about the purpose of the experiment and gave written informed consent to participate in the research. The Bioethics Committee of the Regional Medical Chamber in Gdansk approved the study (KB-10/19).

### 2.3. Instruments

#### 2.3.1. Pain Measurement

The pressure pain threshold (PPT) and pain tolerance (PTOL) tests were conducted using a standard algometer FPN 100 (Wagner Instruments, Greenwich, CT, USA) ranging from 0 to 10 kg, with an attached disc-shaped rubber tip of 1 cm^2^. The algometer measurements were taken on an interval scale to a decimal point; however, the measuring capacity of the device was limited to 10 kg. If a participant could tolerate pressure over 10 kg, the test was stopped and the result was coded as 10.1, the highest possible assumed value for pressure.

Each participant was informed about the meaning of the test and received tips on how to behave during the test. Before the relevant measurement, two attempts were made so that the person tested could distinguish the sensation of pressure from pain and was able to stop the measurement of pressure at the right moment. Pain felt by the participant was manifested by saying “stop”, followed by a recording of the pressure level referred to as the pressure pain threshold (PPT). The test was continued until the subject could not stand the strength of the stimulus and signalled the end of the measurement, which was then classified as the pain tolerance (PTOL). Three measurements were made with increasing pressure (0.5 kg/s = 50 kPa/s) at 10-s intervals on both upper limbs (including the dominant one) at the following locations: on the humerus lateral epicondylus (specification in Table 2 as arm) and on the back of the hand between the thumb and forefinger (specification in Table 2 as hand, the note provides additional information).

#### 2.3.2. Visual Analog Scale (VAS)

The visual analogue scale (VAS) is a standard pain rating scale. After conducting the PPT and PTOL tests, the patients were asked to indicate the pain level from 0–10, where 0 represents “no pain and discomfort” and 10 represents “the worst possible pain and discomfort”.

#### 2.3.3. Pain Control Strategies

The Pain Coping Strategies Questionnaire (CSQ) [24] adapted by Juczyński [25] was employed to assess the main psychological determinants of training effectiveness. The questionnaire contains 42 items describing different pain management strategies, assessed on a Likert scale from 0 (I never do this) to 3 (sometimes I do this) to 6 (I always do this). The CSQ measurement makes it possible to calculate seven different strategies grouped into three general types: cognitive coping (reinterpreting pain sensations, ignoring pain sensations, and declaration of well coping), distraction and substitute activities (distraction and increasing behavioural activity), and catastrophisation (catastrophising, praying/hoping). Each of the strategies included six items. Additionally, two items were analysed independently and scored on a 0–6 scale. Item 43 asked about the feeling of the control (“How much do you feel you are in control of your pain?”) and Item 44 asked about the abilities to decrease experienced pain (“To what extent you feel you are able to reduce your pain?”) [26]. To evaluate the reliability of the measurement, Cronbach’s alpha was used. Among the participants included in this study, the overall reliability was α = 0.94, and for the particular strategies, the values ranged from α = 0.78 to 0.87, indicating a satisfactory level of reliability.

#### 2.3.4. Parenting Styles

Four main parenting styles, according to the classical theories of Baumring (1966) and Maccoby and Martin (1983), were measured retrospectively using subjective survey questions. The participants scored four short characteristics on a scale from 1 (very weak) to 5 (very strong) to assess how strongly they described the atmosphere and the rules at home when they were a child. The instruction was “Please, assess how closely the following characteristics describe the situation at your family home, when you were growing up.”, and the characteristics represented four main parenting styles:**Authoritative**: “There was love, balance, acceptance, and mutual respect”, “The children were involved in making decisions, they had a certain freedom of choice, and there was a lot of conversation”, “Parents raised good examples and arguments”;**Authoritarian**: “Parents were convinced about the power and control over their children, they set strong boundaries, set rules, responsibilities, and made their own decisions”, “The children obeyed for fear of punishment”, “There was no calm and matter-of-fact conversation”;**Permissive**: “There was a relaxed atmosphere and general freedom”; “The parents did not impose anything on their children, did not set any requirements or limits”, “The child could get most of the things he wanted from the parent”;**Rejecting-neglecting**: “Parents’ atmosphere and behaviour at home were variable”; “Linking the previous descriptions”, “The parents acted inconsistently and changed the rules depending on the situation or under the influence of emotions”, “There was no clear division of roles and boundaries”.

### 2.4. Statistical Analysis

Due to the sampling plan, it was only possible to perform a post hoc power analysis. According to the power analysis conducted using the G*Power 3.1 software [26] (for the current single-group model of analysis, which included four variables, two-point measurements with the assumed average correlation between the measurements of 0.50, non-sphericity correction = 1.00, and alpha = 0.95), the current sample size of 85 participants had 0.44, 0.91, 0.99, and 1.00 power to detect an effect size of *f* = 0.10, 0.91, *f* = 0.18, *f* = 0.25, and *f* = 0.40, respectively. It was assumed that *f* = 0.10 indicated a small effect, *f* = 0.18 indicated a small-to-medium effect, *f* = 0.25 indicated a medium effect, and *f* = 0.40 indicated a large effect. However, post hoc power analysis has limitations: its outcome depends on the detected effect size, which may not reflect the true effect size in population [27].

In order to analyse the result of the study, the IBM SPSS Statistics 25 package was used to perform all of the statistical analyses. First, after the cycle of 24 units of physical training, we determined whether there were any significant changes in the pressure pain threshold, pain tolerance, and subjective perception. A repeated-measure ANOVA was applied with two within-subject factors; the model was “2” (body side, left vs. right) × “2” (measurement, the first vs. the second), where the last effect was expected to support the predicted effects of training. The ANOVA was applied separately for the arm and hand, and consequently, for each indicator of pain tolerance. Furthermore, since most of the participants were right-handed, the final version of the analysis accounted for the left and right side of the body (Table 2). In order to interpret the effect size values (η^2^), 0.01–0.05 was considered to be a small effect size, whereas 0.06–0.13 and 0.14–0.20 were considered to be medium and large, respectively. Second, the difference between the second and the first measurements of pain perception was counted to achieve single linear variables to characterise the changes in the three pain indicators after implementing physical training. Then, they were used as the dependent variables in three models of hierarchical linear regression including three groups of possible predictors (i.e., basic sociodemographic factors, pain coping strategies, and parenting styles). Using the change in the *R*^2^ indicator, we determined whenever a new group of possible indicators caused significant changes in the total explained variance. *Z*-scores for all variables were determined before the regression analyses could be combined. The data are stored at https://osf.io/qzhdk/?view_only=f7267578dc11416b9460ad2214a3c92b (accessed on 5 September 2022). The analysis code can be found at https://osf.io/3ukb8/?view_only=30ceb104202047f8851b70293fab6d34 (accessed on 5 September 2022).

## 3. Results

### 3.1. The Effect of Physical Training

#### 3.1.1. Analysis of PPT

The ANOVA calculated for hand pain tolerance did not show the main effect of training (*F*(1, 84) = 2.20, *p* = 0.142, η^2^ = 0.03). The effects of body side and body side x measurement number interaction were not statistically significant (*F*(1, 84) = 2.26, *p* = 0.136, η^2^ = 0.03, and *F*(1, 84) = 2.81, *p* = 0.098, η^2^ = 0.03, respectively). The effect of training was further investigated separately for the left and right sides of the body. As Table 2 indicates, the effect of physical training was statistically significant for the left side of the body, but not for the right side of the body.

The analysis performed on the arm data indicated the effect of training (*F*(1, 84) = 10.57, *p* = 0.002, η^2^ = 0.11). Moreover, the effect of body side and the interaction effect were not statistically significant (*F*(1, 84) = 1.40, *p* = 0.240, η^2^ = 0.02, and *F*(1, 84) = 1.36, *p* = 0.247, η^2^ = 0.01, respectively) (see Table 2 for details). The effect of training was detected for both arms.

#### 3.1.2. Analysis of PTOL

The analysis of the hand data showed that neither the effect of training nor the interaction body side x measurement reached *p* < 0.05, specifically, *F*(1, 84) = 0.09, *p* = 0.769, η^2^ = 0.01 and *F*(1, 84) = 1.92, *p* = 0.169, η^2^ = 0.02; however, the effect of body side was statistically significant (*F*(1, 84) = 6.13, *p* = 0.015, η^2^ = 0.07), specifically, pain tolerance was, in general, higher in the left hand than in the right hand. For pain tolerance in the arm, the ANOVA showed no statistically significant effects (i.e., *F*(1, 84) = 1.64, *p* = 0.204, η^2^ = 0.02 (measurement number), *F*(1, 84) = 1.66, p = 0.201, η^2^ = 0.01 (body side), and *F*(1, 84) = 0.91, *p* = 0.343, η^2^ = 0.01 (body side x measurement number)). Table 2 presents the descriptions.

#### 3.1.3. Analysis of VAS

The ANOVA performed on the hand data showed that the main effect of the physical training and the main effect of the body side were not statistically significant (*F*(1, 84) = 2.20, *p* = 0.142, η^2^ = 0.02 and *F*(1, 84) = 0.32, *p* = 0.573, η^2^ = 0.01, respectively). However, the interaction effect body side x measurement number was statistically significant (*F*(1, 84) = 9.71, *p* = 0.003, η^2^ = 0.10). On one hand, the pattern of the interaction indicated that, at the first measurement, pain tolerance in the left hand was higher than in the right hand (*F*(1, 84) = 4.77, *p* = 0.032, η^2^ = 0.05), whereas the difference between the body sides was not statistically significant when measured after the physical training (*F*(1, 84) = 1.78, *p* = 0.186, η^2^ = 0.02). On the other hand, the training increased the level of pain tolerance for the right hand but not for the left hand (*F*(1, 84) = 5.06, *p* = 0.027, η^2^ = 0.06, and *F*(1, 84) = 0.32, *p* = 0.574, η^2^ = 0.01, respectively).

The analysis of our results related to arms showed that training increased the level of pain tolerance (*F*(1, 84) = 6.22, *p* = 0.015, η^2^ = 0.07). However, the effect of body side was not statistically significant (*F*(1, 84) = 1.72, *p* = 0.194). The interaction effect body side x measurement number was statistically significant *F*(1, 84) = 3.68, *p* = 0.058, η^2^ = 0.04. The pattern of the interaction indicated that, at the first measurement, pain tolerance in the left hand was similar to pain tolerance in the right hand (*F*(1, 84) = 1.11, *p* = 0.295, η^2^ = 0.01), whereas at the second measurement, the pain tolerance in the right arm was statistically higher than in the left arm (*F*(1, 84) = 10.26, *p* = 0.002, η^2^ = 0.11). The effect of training was observed in the right arm but not in the left arm (*F*(1, 84) = 11.51, *p* = 0.001, η^2^ = 0.12, and *F*(1, 84) = 1.89, *p* = 0.173, η^2^ = 0.02, respectively). The descriptions are listed in Table 2.

### 3.2. Strategies of Pain Management among Cardiac Patients

Among every tested strategy, the patients could report a minimum of 0 and a maximum of 33 to 36 points. The most common strategy was a declaration of well coping; the mean result in this scale was *M* = 19.16 (*SD* = 8.91), while the median was 20. The second and third preferable strategies were increasing behavioural activity (*M* = 15.20, *SD* = 8.78, *Me* = 16) and praying/hoping (*M* = 14.39, *SD* = 10.11, *Me* = 13). A very similar intensity was expressed for the strategies of distracting (*M* = 1.82, *SD* = 8.68, *Me* = 11) and ignoring pain sensations (*M* = 12.40, *SD* = 9.02, *Me* = 12). The least preferable strategies were more constructive reinterpretation of the pain (*M* = 7.31, *SD* = 7.52, *Me* = 6) and less constructive catastrophist (*M* = 7.71, *SD* = 7.57, *Me* = 6). Pain coping strategies were independent of the patients’ age and gender.

### 3.3. Strategies of Pain Management in the Context of Psychological Features

The results of the regression analyses explain the first indicator of pain perception (i.e., pressure pain threshold) and are presented in Table 3. Model 1 including only the demographics explained less than 3% of the total variance of PPT, and none of the tested features were a significant predictor of PPT.

After inclusion of the pain coping strategies in Model 2, the explained variance increased to almost 7%, but this change was not statistically significant. One of seven strategies proved to be a significant predictor of changes in PPT (i.e., declaration of well coping with pain). For every additional point on this scale, the changes in PPT after physical training decreased more than 0.30 points. While lower results in the differences between PPT measurements mean a greater decrease in second measurements, these results indicate that patients who declared better abilities to coping with pain experienced a lower pressure pain threshold after the physical training, and therefore a higher sensitivity to pain. Thus, to increase PPT after physical training, a lower attitude to coping with pain only through the cognitive declaration of well coping was more beneficial. The declaration of good pain management was an important predictor, and every 1-point change in this scale caused a decrease in PPT changes of almost 0.40 points. A one-point increase in this assessment corresponded to a decrease in the PPT value of almost 0.30 points. When the results in this assessment went up by one point, the changes in PPT after physical training went down by nearly 0.30 points.

In Model 3, we also included the information about parenting styles in the family home. The explained variance in Model 3 increased to approximately 14% and this change was statistically significant. Only the rejecting–neglecting parenting style was beneficial to pain perception after the physical training. A one-point increase in this parenting style corresponded to a decrease in the PPT value of almost 0.40 points. When the results in this style went up by one point, the changes in PPT after physical training went down by nearly 0.40 points. In addition, controlling the pain was no longer a significant predictor.

This analysis suggests that lower preferences to declaring better coping with pain and growing up in families with less neglecting atmospheres coexist with a greater increase in the pressure pain threshold after the series of physical training among cardiac patients.

Pain tolerance (PTOL), which was only one of the indicators, did not change after the physical training, also becoming the only totally independent one from the tested variables; none of the demographic features, coping strategies as well as parenting styles were significant predictors of changes in the PTOL. Thus, a detailed presentation of the results was omitted.

To explain the possible predictors in changes in subjective pain sensitivity, an analogical regression model is presented (Table 4). While PPT is a more objective indicator of pain sensitivity, VAS is a more subjective indicator and coexists with other possible predictors. In Model 1, this explained 8% of the total variance of changes in VAS, but this variance was statistically non-significant; however, only the patient’s body weight was an important predictor. For each kilogram of body weight gain, the differences in the VAS measurements increased almost 0.30 points; therefore, heavier patients experienced a greater decrease in the subjective feeling of pain. This effect remained significant across all of the tested models.

After including the pain coping strategies as predictors (Model 2), the relationship between the body weight and change in subjective pain weakened but was significant, while the scope of the explained variance increased significantly by 23%. Two of the coping strategies were found to be important (i.e., prayer/hope decreased the changes in VAS by 0.30 and ignoring pain increased the changes in VAS by 0.40 points, respectively.

In this case, contrary to the changes in PPT, we expected to observe a lower intensity in VAS after the physical training, represented by negative changes between the measurements. Therefore, in this case, negative regression coefficients are consistent with the hypothesis of a decrease in subjective feelings of pain. The results suggest that the patients who were more likely to pray and be hopeful, indeed, declared lower intensity in pain after the physical training. However, this relationship was reversed for ignoring the pain. These two predictors remained significant after enriching the model by the parenting styles in Model 3. The total explained variance increased by 3%, but this change was not statistically significant. In this case, parenting styles were independent of subjective changes in pain; therefore, the interpretation of the final results did not differ between Models 2 and 3.

If required, the zero-order correlations between the variables tested in the regression models can be generated based on this analysis code: https://osf.io/3ukb8/?view_only=30ceb104202047f8851b70293fab6d34 (accessed on 5 September 2022).

## 4. Discussion

Cardiac rehabilitation (CR) is an integrated treatment process that focuses on providing supervised aerobic exercise training as well as additional support, especially promoting a healthy lifestyle, dietary recommendations, and psychosocial support to patients with cardiovascular disease (CVD). Cardiac rehabilitation has been shown to reduce the total mortality by 20% and cardiac mortality by 26.5% [28]. Exercise-based rehabilitation reduces hospital readmissions after cardiovascular events, improves mental health, and is able to promote neuroplasticity and neuroprotection by acting at the cytokine and hormonal level [29]. Moreover, the results of previous studies have unequivocally indicated that regular physical activity is a key link in the “process” of shaping sensitivity to pain by modulating the threshold of receiving both painful and non-painful stimuli as well as pain tolerance [18,30]. However, the literature lacks clear guidelines for quantitative and qualitative body loads that can modulate the perception of pain.

Pain is a subjective phenomenon, the informative effect of which is shaped by the currently perceived nociceptive stimuli, and it is modulated by mood, attitude to life, and past experiences. This research aimed at taking into consideration and describing these interacting aspects including the influence of physical training on pressure pain threshold and pain tolerance as well as subjective pain perception and the possible modulating effect of the way of coping with stress and parenting styles among patients who participated in a cardiac rehabilitation course.

Our results indicate that cardiac rehabilitation affects the perception of pain. Patients participating in the described program achieved significantly higher pressure pain threshold values measured on an arm and hand. No differences occurred in pain tolerance on both upper limbs. The data from previous studies on healthy people and athletes have emphasized the significant influence of physical activity on the range of pain tolerance and less influence on the pain threshold [31,32]. According to the studies, the pressure pain threshold is relatively constant and is less influenced by psychological and psychosocial factors [33,34]; however, the available pressure pain threshold data do not present a uniform picture. In the studied group, physical training showed a significant impact on the pressure pain threshold, except for the dominant hand. It may be suggested that in this case, regular stimulation of the nondominant limb and right arm by physical exercise increased the pain threshold. It is known that the sensory system is structurally and functionally associated with the motor system. Therefore, treatments aiming at increasing motor control can be used for pain control [35,36]. Additionally, the research results confirmed that the cortical structures (i.e., the primary motor cortex (M1)) can significantly reduce pain [37].

Regarding pain tolerance, we found a lack of differences after a cycle of 24 workouts, which can be explained by the level of physical activity and its regularity. Each patient had an individually determined amount of load according to the results of the stress test at the time of qualification for the rehabilitation. In addition, despite having the recommended frequency of exercise (i.e., three times a week), the patients were not able to meet the recommendation. The reason for this could be an individual level of kinesiophobia, which may negatively affect the cardiac rehabilitation process [38]. The results of the subjective pain assessment (measured by VAS) in the studied patients were remarkable. After the rehabilitation, the subjective pain sensation (VAS) after the PPT and PTOL tests showed that the patients reported a higher subjective pain sensation (measured by VAS) than at the beginning of rehabilitation (significant differences related to the dominant limb).

The regression analyses performed indicated that one of the factors may be the parenting style in the family home. People who grew up in a neglected atmosphere after completing rehabilitation had an increased pain threshold after training. Deficient parental–child interactions create vulnerabilities that disrupt psychological functioning and stress responsive physiological systems load [39]. This has also been confirmed in another study. According to the authors of one study, children raised in cold, unsupportive, and/or neglectful family environments may have greater difficulty in controlling their emotions and are more sensitive to threat cues than other children [40]. One of the symptoms of a mental reaction to pain is anxiety. Pain causes anxiety, and fear, in turn, makes it more painful (reinforces pain). The parenting style significantly determines the method of coping with stress (pain).

Pain coping strategies are active forms undertaken by the patient in pain situations, and their effectiveness depends on the stressful situation and the extent to which it is controlled by the individual. Patients who were more likely to use prayers and who were more hopeful declared lower intensity in pain (VAS) after the physical training. Perhaps the significantly more frequent use of prayer in coping with pain is related to the age of the respondents. This was supported by the results of research conducted among patients with coxarthrosis disease [41]. On the other hand, the patients who declared stronger ignoring pain tendency as one of the coping strategies displayed an increased intensity in pain after the training. This suggests that ignoring pain may not be an effective coping style in controlling pain. In addition, higher body weight was related to increased intensity in pain after the training. This finding is consistent with that of previous findings [42].

In conclusion, in therapy with cardiac patients, the function of stress factors including pain and anxiety on the quality of life should be considered. Some researchers recommend that a psychologist train cardiac patients in coping strategies, together with non-invasive and invasive medical treatments as well as physical exercise in order to improve physical and cognitive recovery [43,44].

## 5. Limitations

The authors are aware of certain limitations resulting from the adopted methodology and access to the patients. One limitation was the patients’ freedom to perform training sessions two, three, or five times a week; however, each of them conducted the assumed 24 workouts. The advantage of this system, however, is that the frequency of exercise can be adjusted to the patients’ individual subjective feeling of being able to exercise. The lack of the analysis of patients with regard to previous surgery or diagnosed cardiovascular events such as myocardial infarction, stent implantation, cardiac surgery, or coronary disease could also be considered as a limiting factor. This was undertaken to avoid a small group size, recognizing that the type of surgery or the extent of the procedure may influence the qualitative or quantitative aspects of pain perception. Furthermore, even in a homogeneous group of MI patients, one is dealing with varying degrees of myocardial damage and the consequences of myocardial infarction.

## 6. Conclusions

In summary, our data suggest that regular exercise is associated with specific changes in the pressure pain threshold, but an association with changes in pain tolerance cannot be confirmed. It is likely, and this requires further research, that stronger physical stimuli such as intense exercise that must be performed regularly are required to produce significant changes in pain tolerance. Lower scores on the “explanation of good coping” strategy and growing up in families with a less neglectful atmosphere were associated with higher pressure pain thresholds after a series of physical training sessions in cardiac patients. Patients with active behaviours and those who used a prayer/hope strategy to cope with pain reported lower pain scores after exercise. However, ignoring pain and increased body weight could compromise the potential benefits of such physical training.

## Figures and Tables

**Table 1 ijerph-19-11276-t001:** The final sample description (*N* = 85).

Variable	M	SD	Min.	Max.	Frequency
Age	65.36	8.64	37	84	–
Gender	–	–	–	–	Females, 31 (36.47%) and males, 54 (63.53%)
BMI	29.13	4.52	21.80	42.93	–
Dominant hand	–	–	–	–	Right-handed, 83 (95.29%) and left-handed, 2 (2.35%)

**Table 2 ijerph-19-11276-t002:** The effect of the physical training (*N* = 85).

MN	Pressure Pain Threshold (PPT)				Pain Tolerance (PTOL)				Subjective Pain (VAS)			
	L Hand	R Hand	L Arm	R Arm	L Hand	R Hand	L Arm	R Arm	L Hand	R Hand	L Arm	R Arm
1	1.36 (0.29)	1.35 (0.25)	1.36 (0.31)	1.31 (0.23)	2.22 (0.81)	2.09 (0.87)	2.60 (1.18)	2.58 (1.07)	3.95 (2.42)	3.68 (2.06)	3.34 (2.03)	3.22 (2.10)
2	1.42 (0.34)	1.36 (0.23)	1.42 (0.32)	1.41 (0.27)	2.15 (0.75)	2.12 (0.77)	2.87 (2.08)	2.65 (0.90)	4.09 (2.34)	4.20 (2.15)	3.66 (2.05)	3.98 (2.02)
*F*(1, 84)	**3.66**	0.17	**3.98**	**13.43**	0.92	0.17	1.54	0.52	0.32	**5.06**	2.99	**11.51**
*p*	**0.024**	0685	**0.049**	**<0.001**	0.340	0.685	0.217	0.475	0.574	**0.027**	0.173	**0.001**
η^2^	**0.05**	0.01	**0.05**	**0.14**	0.01	<0.01	0.02	0.01	0.01	**0.06**	0.02	**0.12**

Note: MN, measurement number; L, left side; R, right side. Standard deviations are provided in the parentheses. The values in bold indicate statistically significant differences between the measurements (*p* < 0.05).

**Table 3 ijerph-19-11276-t003:** Hierarchal linear regression predicting changes in the pressure pain threshold (PPT) after physical training.

Predictors of Changes in PPT		Model 1, *R*^2^ = 0.03 Δ*F*(4, 80) = 0.51, *p* = 0.729			Model 2, *R*^2^ = 0.15 Δ*F*(9, 71) = 1.12, *p* = 0.360			Model 3, *R*^2^ = 0.29 Δ*F*(4, 67) = 3.38, *p* = 0.014		
		ß	*t*	*p*	ß	*t*	*p*	ß	*t*	*p*
Sociodemographics	Age	−0.11	−0.94	0.352	−0.04	−0.328	0.744	−0.03	−0.22	0.826
	Body height	−0.12	−0.63	0.528	−0.02	−0.12	0.905	−0.05	−0.25	0.803
	Body weight	0.13	1.02	0.309	0.02	0.12	0.904	0.03	0.24	0.813
	Gender	−0.09	−0.55	0.583	−0.05	−0.28	0.785	0.04	0.23	0.823
Coping with pain strategies	Distracting				−0.09	−0.42	0.677	−0.01	−0.03	0.974
	Reinterpreting the pain				0.20	1.06	0.291	0.12	0.60	0.551
	Catastrophising				0.11	0.79	0.435	0.10	0.77	0.443
	Ignoring the pain				0.10	0.49	0.625	0.29	1.45	0.152
	Praying and hoping				−0.15	−0.98	0.331	−0.12	−0.84	0.404
	Declaration of well coping				**−0.34**	**−1.99**	**0.050**	−**0.42**	−**2.55**	**0.013**
	Increasing behavioural activity				0.24	1.04	0.302	0.13	0.57	0.573
	Controlling the pain				**−0.28**	**−2.01**	**0.039**	−0.21	−1.65	0.104
	Decreasing the pain				0.02	0.09	0.926	0.05	0.25	0.801
Parenting Styles	Authoritative							−0.22	−1.71	0.092
	Authoritarian							−0.06	−0.44	0.660
	Permissive							−0.10	−0.81	0.419
	**Rejecting-Neglecting**							−**0.35**	−**2.73**	**0.008**

Note: The values in bold indicate statistically significant differences between the measurements (*p* < 0.05).

**Table 4 ijerph-19-11276-t004:** Hierarchical linear regression predicting changes in subjective pain (VAS) after physical training.

Predictors of Changes in VAS		Model 1, *R*^2^ = 0.08 Δ*F*(4, 80) = 1.78, *p* = 0.140			Model 2, *R*^2^ = 0.28 Δ*F*(9, 71) = 2.63, *p* = 0.011			Model 3, *R*^2^ = 0.31 Δ*F*(4, 67) = 0.80, *p* = 0.530		
		ß	*t*	*p*	ß	*t*	*p*	ß	*t*	*p*
Sociodemographics	Age	−0.04	−0.34	0.733	0.08	0.68	0.501	0.07	0.60	0.553
	Body high	−0.32	−1.77	0.080	−0.29	−1.64	0.105	−0.37	−1.92	0.060
	Body weight	**0.27**	**2.22**	**0.029**	**0.24**	**2.01**	**0.049**	**0.26**	**2.15**	**0.035**
	Gender	−0.28	−1.70	0.094	−0.25	−1.51	0.135	−0.27	−1.51	0.137
Coping with pain strategies	Distracting				0.29	1.44	0.156	0.32	1.57	0.121
	Reinterpreting the pain				−0.16	−0.90	0.371	−0.24	−1.29	0.203
	Catastrophising				0.06	0.50	0.619	0.05	0.41	0.683
	Ignoring the pain				**0.37**	**2.06**	**0.043**	**0.46**	**2.44**	**0.017**
	Praying and hoping				−**0.32**	−**2.32**	**0.023**	−**0.30**	−**2.14**	**0.040**
	Declaration of well coping				−0.09	−0.59	0.561	−0.13	−0.84	0.406
	Increasing behavioural activity				−0.18	−0.85	0.398	−0.16	−0.75	0.457
	Controlling the pain				−0.18	−1.46	0.149	−0.17	−1.40	0.168
	Decreasing the pain				−0.32	−1.81	0.075	−0.36	−1.94	0.057
Parenting Styles	Authoritative							0.04	0.31	0.756
	Authoritarian							0.05	0.43	0.672
	Permissive							−0.08	−0.66	0.509
	**Rejecting-Neglecting**							−0.16	−1.30	0.199

Note: The values in bold indicate statistically significant differences between the measurements (*p* < 0.05).

## Data Availability

The data are stored at https://osf.io/qzhdk/?view_only=f7267578dc11416b9460ad2214a3c92b (accessed on 5 September 2022). The analysis code can be found at https://osf.io/3ukb8/?view_only=30ceb104202047f8851b70293fab6d34 (accessed on 5 September 2022). The data can also be made available from the corresponding author (K.L.) upon reasonable request.

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
