# Peer review of "Is Physical Activity an Effective Factor for Modulating Pressure Pain Threshold and Pain Tolerance after Cardiovascular Incidents?"

_ijerph, 2022, doi:10.3390/ijerph191811276_

Round 1

Reviewer 1 Report (Previous Reviewer 2)

The article by Katarzyna Leznicka et al. “Is Physical Activity an Effective Factor for Modulating Pressure Pain Threshold and Pain Tolerance after Cardiovascular Incidents?” evaluates the differences in pain perception and tolerance in cardiac patients during rehabilitation. The Authors demonstrated that patients achieved significantly higher PPT levels after cardiac rehabilitation course, but did not differ in PTOL. Moreover, Authors showed an association between pain coping strategies and parenting styles with PPT. For some of the parameters, the differences affected only one of the parties examined (not bilaterally).

Comments:

Lines 28-31 Abstract does not contain any numerical results. There is only an interpretation of the results. That should be corrected.

Lines 33-34 […] a stronger and regular stimulus in the form intense and regular physical exercise is needed to induce changes in pain tolerance.” This conclusion does not come from the study, it is only a speculation. The same in Lines 439-441

Lines 131-136 I strongly suggest that Table 2 should be included in the results section and Lines 131-133 should be modified.

Line 181: Amend “Participation in the study required written consent from each of the patients.” into “The participants were informed about the purpose of the experiment and gave written informed consent to participate in the research. The Bioethics Committee of the Regional Medical Chamber in Gdansk approved the study (KB -10/19).” (from Lines 198-200).

 “Study Design” should be placed first in the methods section (e.g. 2.1).

 “Power Analysis” should be moved to “Statistical Analysis”

 The Tables: All abbreviations should be explained in the captions.

Table 2 is not clear and should be rebuilt.

In Tables 2 and 3: Significant differences are marked in bold (?). Please explain in the captions.

In Table 3:  the word “Coping” is lost

Main concern:

Pain is an important symptom for patients and healthcare professionals to identify and assess the severity of the disease. Patients with asymptomatic myocardial ischemia are known to have much higher pain thresholds. As I understand it, the results of this study suggest that cardiac rehabilitation leads to higher pressure pain threshold (PPT) and does not affect PTOL. Higher pain thresholds are likely to increase the number of undetected heart attacks. The results are therefore difficult to interpret from a practical point of view and are not sufficiently discussed in the paper. In the discussion, I found no explanation why the differences were not bilateral. The authors did not consider any comorbidities (e. g. diabetes, depression, smoking) that could be confounding factors. The manuscript does not contain any information about the course of rehabilitation. The practical implication of the results and its scientific significance is not highlighted in the paper. The text needs substantial corrections.

Author Response

Reviewer 1

Thank you for raising these very important comments and concerns.

Response to comments:

Lines 28-31 Abstract does not contain any numerical results. There is only an interpretation of the results. That should be corrected.

The requested information is now added. We focused on effect sizes and p-values.

Lines 33-34 […] a stronger and regular stimulus in the form intense and regular physical exercise is needed to induce changes in pain tolerance.” This conclusion does not come from the study, it is only a speculation. The same in Lines 439-441

A very valuable remark. Speculation has been removed from the abstract and we have underlined in the text that this area requires further research, as well.

Lines 131-136 I strongly suggest that Table 2 should be included in the results section and Lines 131-133 should be modified.

The table was moved to the results section.

Line 181: Amend “Participation in the study required written consent from each of the patients.” into “The participants were informed about the purpose of the experiment and gave written informed consent to participate in the research. The Bioethics Committee of the Regional Medical Chamber in Gdansk approved the study (KB -10/19).” (from Lines 198-200).

The suggested change has been incorporated.

 “Study Design” should be placed first in the methods section (e.g. 2.1).

 “Power Analysis” should be moved to “Statistical Analysis”

 The Tables: All abbreviations should be explained in the captions.

Table 2 is not clear and should be rebuilt.

In Tables 2 and 3: Significant differences are marked in bold (?). Please explain in the captions.

In Table 3:  the word “Coping” is lost

The study’s design was moved and now is presented after the participant section.

Power analysis is now a part of the statistical analysis section.

We checked the tables’ captions and notes, suitable corrections have been implemented. We hope that the communication is now clearer than before.

Indeed, Table 2 includes a large portion of information. We consider that remaking Table 2 would require to split the data info three tables to make the communication more transparent; however, for the sake of conciseness, we decided to keep it as it is.

Table 4 was corrected by adding “Declaration of Well Coping”.

Main concern:

Pain is an important symptom for patients and healthcare professionals to identify and assess the severity of the disease. Patients with asymptomatic myocardial ischemia are known to have much higher pain thresholds. As I understand it, the results of this study suggest that cardiac rehabilitation leads to higher pressure pain threshold (PPT) and does not affect PTOL. Higher pain thresholds are likely to increase the number of undetected heart attacks. The results are therefore difficult to interpret from a practical point of view and are not sufficiently discussed in the paper. In the discussion, I found no explanation why the differences were not bilateral. The authors did not consider any comorbidities (e. g. diabetes, depression, smoking) that could be confounding factors. The manuscript does not contain any information about the course of rehabilitation. The practical implication of the results and its scientific significance is not highlighted in the paper. The text needs substantial corrections.

The explanation for the differences in the PPT between the limbs was suggested in the sentence ( line...) “In the studied group, physical training showed a significant impact on the pressure pain threshold, except for the dominant hand. One can speculate that  in this case, a regular stimulation non-dominant limb and right arm  done through physical exercise increased the pain threshold.”

The non-dominant limb and the right arm are generally less involved in performing simple everyday activities, hence we can assume, that systematic stimulation in the form of physical exercise or current, everyday sensory events could affect the differences in PPT.

All data on comorbidities, lifestyle, and medications taken were collected at the time of study recruitment. In additional analyses, we could not control for the random factors suggested in the review (low number samples). Thus, the robustness of the reported effects cannot be challenged through control of such additional factors. Future studies could control for factors that can help assess the robustness of such an intervention, e.g., diabetes, smoking, depression.

In the discussion and conclusion, we note and indicate that physical activity, "well coping" declarations, and familial neglect can modulate pain perception in cardiac patients.

The information about rehabilitation course was added in part 2.1. Study Population: The second stage of cardiac rehabilitation consists of 24 training units carried out from 1 to 5 days a week, according to the patient's abilities. Every training unit included:

- interval training on a bicycle ergometer of 30 minutes with 3-minute load periods and alternating with 2-3-minute rest periods,

- general fitness exercises, including breathing exercises, relaxing and stretching lasting 30 minutes.

We thank you for all your suggestions, which we have improved in the manuscript, some of which we have formulated in response to you.

Reviewer 2 Report (Previous Reviewer 1)

Introduction: The authors state the aim of the study clearly at the end of the introduction, though the last and second to last paragraphs are redundant. I also think that the first part of the introduction discussing silent myocardial infarctions and cardiovascular mortality may be misleading regarding the overall aim of the paper. Please consider removing this part of the introduction. I think the benefit of studying pain measurements in patients that are undergoing cardiac rehab is the relatively uniform physical therapy each patient undergoes, leading to more consistency in pain measurements.

Methods: The measurements and scales are described well. 

Results: The result text is relatively hard to follow. Please consider only including p-values to represent statistical significance. Please consider including a key for Table 2 so that the reader does not have to turn to the methods to reference what each number means. 

Discussion: I think the authors make a fair conclusion that physical activity, "well coping" declarations, and familial neglect can modulate pain perception in patients. I would caution them against stating that cardiac rehabilitation can reduce recurrence of cardiac events, as I do not see this association made in this study (line 421). 

Author Response

Reviewer 2

Thank you for raising these very important comments and concerns.

Introduction: The authors state the aim of the study clearly at the end of the introduction, though the last and second to last paragraphs are redundant. I also think that the first part of the introduction discussing silent myocardial infarctions and cardiovascular mortality may be misleading regarding the overall aim of the paper. Please consider removing this part of the introduction. I think the benefit of studying pain measurements in patients that are undergoing cardiac rehab is the relatively uniform physical therapy each patient undergoes, leading to more consistency in pain measurements.

Thank You for a very valuable remark. We reduced suggested paragraphs (lines 88-92) in introduction, recreated first part and removed some not necessary items connected with silent myocardial infarctions.

Methods: The measurements and scales are described well. 

Thank you very much for this positive remark.

Results: The result text is relatively hard to follow. Please consider only including p-values to represent statistical significance. Please consider including a key for Table 2 so that the reader does not have to turn to the methods to reference what each number means. 

Thank you for the suggestions. Table 2 was corrected by adding additional information about the tested variables. Besides, we refer the reader to the details provided in the table’s note to ensure that the table is self-explanatory. Also, we corrected the description of the analysis of the pain management strategies. All the essential statistics were moved to the tables. In consequences of all the changes, the tables (2, 3, 4) contain information essential for the study’s goals. We hope that the result section is now easier to follow.

We decided to present effect sizes along with p-values in order represent the meaningfulness of the results. Knowing the effect sizes can be helpful to determine the impact of the training and thus can be useful in determining possible future similar interventions.

Discussion: I think the authors make a fair conclusion that physical activity, "well coping" declarations, and familial neglect can modulate pain perception in patients. I would caution them against stating that cardiac rehabilitation can reduce recurrence of cardiac events, as I do not see this association made in this study (line 421). 

As suggested, we have removed the statement about the reduction of recurrences of cardiac events, although this particular sentence refers to the literature and not to our research.

Reviewer 3 Report (New Reviewer)

Dear authors,

Your paper seems very interesting but some concerns have to be addressed.

The method section has to be integrated in order to reinforced in a weak point, that is the survey. Although it is understandable why You choos it, it is necessary to find other similar uses in literature with regard to pain and physical activity. To do that I suggest the following references:

- Farì G, Fischetti F, Zonno A, Marra F, Maglie A, Bianchi FP, Messina G, Ranieri M, Megna M. Musculoskeletal Pain in Gymnasts: A Retrospective Analysis on a Cohort of Professional Athletes. Int J Environ Res Public Health. 2021 May 20;18(10):5460. doi: 10.3390/ijerph18105460. PMID: 34065250; PMCID: PMC8160814.

The discussion needs to be improved. You only marginally explored the theme of the importance of physical activity from a rehabilitative point of view. A further paragraph is desirable in order to highlight the effects of physical activity in the rehabilitation project of other pathology, also with regard to pain. To do that I suggest the following references:

- Farì G, Lunetti P, Pignatelli G, Raele MV, Cera A, Mintrone G, Ranieri M, Megna M, Capobianco L. The Effect of Physical Exercise on Cognitive Impairment in Neurodegenerative Disease: From Pathophysiology to Clinical and Rehabilitative Aspects. Int J Mol Sci. 2021 Oct 27;22(21):11632. doi: 10.3390/ijms222111632. PMID: 34769062; PMCID: PMC8583932.

- Ekiz T, Kara M, Ata AM, Ricci V, Kara Ö, Özcan F, Özçakar L. Rewinding sarcopenia: a narrative review on the renin-angiotensin system. Aging Clin Exp Res. 2021 Sep;33(9):2379-2392. doi: 10.1007/s40520-020-01761-3. Epub 2021 Jan 4. PMID: 33394457.

Finally, please check the general layout before resedning the manuscript file.

Best regards and good luck

Author Response

Reviewer 3

Thank you for raising these very important comments and concerns.

The method section has to be integrated in order to reinforced in a weak point, that is the survey. Although it is understandable why You choos it, it is necessary to find other similar uses in literature with regard to pain and physical activity.

To do that I suggest the following references:

Following your suggestion we rebuilt and integrated method section based on your references. It was not so easy because methodology used in this literature was a little different from ours eg. direct examination vs on line questionnaire but suggestions about details were very helpful. It was also very difficult to compare the examined populations (young vs. aged).

- Farì G, Fischetti F, Zonno A, Marra F, Maglie A, Bianchi FP, Messina G, Ranieri M, Megna M. Musculoskeletal Pain in Gymnasts: A Retrospective Analysis on a Cohort of Professional Athletes. Int J Environ Res Public Health. 2021 May 20;18(10):5460. doi: 10.3390/ijerph18105460. PMID: 34065250; PMCID: PMC8160814.

The discussion needs to be improved. You only marginally explored the theme of the importance of physical activity from a rehabilitative point of view. A further paragraph is desirable in order to highlight the effects of physical activity in the rehabilitation project of other pathology, also with regard to pain. To do that I suggest the following references:

- Farì G, Lunetti P, Pignatelli G, Raele MV, Cera A, Mintrone G, Ranieri M, Megna M, Capobianco L. The Effect of Physical Exercise on Cognitive Impairment in Neurodegenerative Disease: From Pathophysiology to Clinical and Rehabilitative Aspects. Int J Mol Sci. 2021 Oct 27;22(21):11632. doi: 10.3390/ijms222111632. PMID: 34769062; PMCID: PMC8583932.

That was just to the point. Absolutely important suggestion. Physical exercise, which is able to promote neuroplasticity and neuroprotection in aged people is additional positive factor which should be accentuated. So we did in our discussion.

- Ekiz T, Kara M, Ata AM, Ricci V, Kara Ö, Özcan F, Özçakar L. Rewinding sarcopenia: a narrative review on the renin-angiotensin system. Aging Clin Exp Res. 2021 Sep;33(9):2379-2392. doi: 10.1007/s40520-020-01761-3. Epub 2021 Jan 4. PMID: 33394457.

We do not mention RAS in our work because of size of the study. But this is an inspiration for the future work just to find and check influence of this system on pain perception especially in cardiac patient with ACEI treatment ( we do not have this information in our group ). On the other hand it was for us very important to put this reference in our work to show importance  of physical exercise on holistic recovery in rehabilitation ( pain, cognitive and physical functions ).

Finally, please check the general layout before resedning the manuscript file.

We try to do our best, but some aspects are in hand of our editor.

Best regards and good luck

Thank You very much. The same to you.

Reviewer 4 Report (New Reviewer)

This is an important study to help cardio rehab patients address pain. Well done. The rationale for the study is well supported by the cited literature and by the results of the study. I've included a pdf file with a few minor comments about the text.

Author Response

Reviewer 4

This is an important study to help cardio rehab patients address pain. Well done. The rationale for the study is well supported by the cited literature and by the results of the study. I've included a pdf file with a few minor comments about the text.

Thank you very much for the positive review and kind words. All the  comments from the pdf were included in the text, they helped us a lot.

Reviewer 5 Report (New Reviewer)

Dear

First of all, thank you for submitting this application to IJERPH.

I've been thinking a lot about this document. However, I have doubts about science and causality.

The “pressure pain threshold” is a very important variable in this study.

1. Why was pressure measured on cardiac patients?

This is because "pressure" and "chest pain" felt by heart patients are different.

The pressure pain threshold using the “algometer FPN 100” and the “chest pain” of cardiac patients are not related at all. And it has absolutely nothing to do with the clinical aspect of the heart.

And, in the introduction, the word "pressure" first appears on line 85. The author does not sufficiently explain the meaning of pressure measurement to cardiac patients in the introduction.

2. Furthermore, what is the significance of pressure measurements on the hands and arms of cardiac patients?

It's not described anywhere.

3. In the introduction, there is information about the participation rate in cardiac rehabilitation. However, the study design is very weak about the participation rate in cardiac rehabilitation.

Unfortunately, I have the opinion that it is unsuitable for publication.

Author Response

Reviewer 5

Thank you very much for your remarks and comments. They all were very useful for us. We rethought the concept of our work and made very important changes to follow your recommendations. Please look at the manuscript once more and we hope that with our answers, you can find below, this new work will be for you more satisfactory.

  1. Why was pressure measured on cardiac patients?

This is because "pressure" and "chest pain" felt by heart patients are different.

The reviewer is right in pointing out that "pressure" and "chest pain" have different origin. However, the neurophysiological basis of pain is common. Pain is a sensory phenomenon that is highly individualised, as it involves the physical, cognitive and emotional spheres of human life and does not have to be a response to tissue damage, but according to its definition, it can also manifest "in relation to potential tissue damage or sensations which may be described using terms referring to such damage”. Physiologically, pain has an informative and warning function, as it signalises direct or potential tissue damage. It also serves as a protective mechanism by significantly increasing the chances of human survival. Furthermore, it allows for learning about the environment and avoiding situations that may be hazardous to one's health or life.

Pain is a complex phenomenon. In addition to its most characteristic sensory-discriminative aspect that allows for locating the stimulus and evaluating its intensity, pain also has emotional (affective), locomotor and autonomic (vegetative) aspects. In the clinical aspect, pain serves a diagnostic function, indicating the site of an injury and ongoing pathological processes, as well as a monitoring function, as increasing or decreasing pain allows for the evaluation of a given condition. Pain or lack thereof is also a subjective criterion of disease or good health, respectively.

The pressure pain threshold using the “algometer FPN 100” and the “chest pain” of cardiac patients are not related at all. And it has absolutely nothing to do with the clinical aspect of the heart.

The Wagner Pain Test™ Model FPK Algometer is the worldwide preferred threshold and pain tolerance gage using by scientists, physicians, and other health experts. Pressure algometry has been proven a valid measure of pain of various etiologies. Moreover, the usefulness of the pressure algometer in the diagnosis and treatment of different diseases has been described in many scientific publications for example:

Ebru Kaya Mutlu,,  Arzu Razak Ozdincler. Reliability and responsiveness of algometry for measuring pressure pain threshold in patients with knee osteoarthritis. J Phys Ther Sci. 2015; 27(6): 1961–1965.

Lisbeth Hven, Poul Frost, Jens Peter Ellekilde Bonde. Evaluation of Pressure Pain Threshold as a Measure of Perceived Stress and High Job Strain. Biomed Res Int. 2015; 2015: 575038.  doi: 10.1155/2015/575038.

Natasha Bergmann et al. The Effect of Daily Self-Measurement of Pressure Pain Sensitivity Followed by Acupressure on Depression and Quality of Life versus Treatment as Usual in Ischemic Heart Disease: A Randomized Clinical Trial. PLOS ONE; https://doi.org/10.1371/journal.pone.0097553.

Sadigh Bita et al. β-Endorphin Modulates Adenosine Provoked Chest Pain in Men, But Not in Women—A Comparison Between Patients With Ischemic Heart Disease and Healthy Volunteers. The Clinical Journal of Pain. 2007; 23 (9): 750-755.

And, in the introduction, the word "pressure" first appears on line 85. The author does not sufficiently explain the meaning of pressure measurement to cardiac patients in the introduction.

In our research on cardiac patients, the algometer performed the function of a standardized mechanical stimulus applied to patients to determine the pain threshold and pain tolerance.

Pain, as already mentioned, is a subjective feeling and may be subject to strengthening or inhibition of nociceptive processes in central nerve system. The application of a standardized stimuli, in this case a mechanical ones, makes it possible to recognize and quantify such processes. This also applies to cardiac patients.

  1. Furthermore, what is the significance of pressure measurements on the hands and arms of cardiac patients?

It's not described anywhere.

The accuracy with which a tactile stimulus can be detected depends on the area of the body. Differences in sensitivity to harmful stimuli depend on the distribution of receptors and their density, therefore two different places were selected: the arm and the hand. We did not include it in the manuscript the brighter explanation because it is the issues of human physiology.

  1. In the introduction, there is information about the participation rate in cardiac rehabilitation. However, the study design is very weak about the participation rate in cardiac rehabilitation.

The final size of the group results from many aspects:

- patient examinations were conducted in a rehabilitation hospital where the patients came in person only for cardiac training, they were not hospitalized,

- during the 6-month recruitment, over 40% of patients did not agree to participate in the study,

- the attrition rate was relatively low (less than 8%). Besides, the statistical tests we used accounted for available degrees of freedom and thus robust given the study’s design and tested relationships.

Once more thank you for your time, engagement and help. 

Round 2

Reviewer 1 Report (Previous Reviewer 2)

The Authors addressed most of my previous comments and improve the quality of the present manuscript. The article is much clearer.

The abstract summarizes the main points. However, I don’t know what “ps” means in Lines 29-30: “The main results of the study showed that patients achieved significantly higher pressure pain thresholds after a physical training cycle (ps < 0.05, 2 = 0.05-0.14), but found no differences in pain tolerance (ps > 0.05).” In addition “2 = 0.05-0.14” is also completely not clear for me.

The aim of the study is stated clearly and the introduction gives enough background.

Materials and methods are used appropriate and are adequately described.

The results are presented clear.

Discussion and conclusions are consistent with the results.

Main finding are summarized and put into the context.

Limitations are sufficiently discussed.

The references are adequate.

Author Response

As suggested, we have added numerical information to the abstract. Unfortunately, in the process of text processing, essential elements of the addition were lost. We thank you for noticing this error, and please note that the current version of the abstract contains the correct symbols. Notably, we used “ps” as an abbreviation for “p-values”. It was necessary because the described effects account for multiple comparisons and thus “ps < 0.05” or “ps > 0.05” is the adequate technique to annotate the differences.

Reviewer 5 Report (New Reviewer)

You did a lot of work to make changes.

A more natural English expression is recommended.

Author Response

The manuscript was corrected by two proofreading and we rely on their edits. We attached a certificate to evidence the professional text editing.

This manuscript is a resubmission of an earlier submission. The following is a list of the peer review reports and author responses from that submission.

Round 1

Reviewer 1 Report

The study's stated goal was to evaluate the effect of physical exercises on pain threshold and tolerance, and modulating effects of other factors in patients participating in a cardiac rehabilitation course. I think my main concern is that I am still not sure why evaluating pain threshold and tolerance is of significance in patients in a cardiac rehabilitation course. Will this affect their success in participating in a cardiac rehabilitation course? I think evidence to support why we should study pain tolerance / threshold in the cardiac rehab population could be made clearer and supported in a stronger fashion in the introduction. 

There is a lot of detail in the methods, and yet this was relatively difficult to follow. I wonder if the study design could be placed a little earlier in the methods section. I think some of the results are in the methods section (e.g. describing the study population and first reference of table 2), which may be better read in the results section. 

For the results section, I think the authors could take out the statistical numbers aside from the p-value, and focus on the verbal interpretation of the results. 

What is the significance of why the pain threshold was different on one hand and not the other for the participants? More discussion of why addressing pain tolerance and threshold in cardiac patients could be emphasized as well in the conclusions section. 

Overall there are moderate English edits needed throughout the manuscript, especially in the abstract. 

Reviewer 2 Report

The article by Katarzyna Leźnicka et al. entitled “The impact of physical activity on pain perception among cardiac patients: the importance of pain coping strategies” is a study, aimed to assess the differences in pain perception and tolerance in cardiac patients during rehabilitation.

Differences in pain perception have been a topic of increased interest in recent years. The novel contribution of the paper and clinical implication need to be highlighted in the paper.

English language requires major improvements, the text requires many corrections, the tables require more care.

Major comments:

Abstract: The number of included (n=85) should be specified. The Authors do not give any results, only their interpretation.

Methods: “The studied group with a history of myocardial infarction (MI), coronary angioplasty and cardiac surgery included….”  You mean that only patients with MI + PCI + CABG were included ?

A brief description of the study sample (baseline demographic and clinical characteristics) would be welcomed in Table 1. What about the prevalence of diabetes and other comorbidities in the study group? It can have a significant connection to the pain perception.

Why is Table 2 included in the Methods section if it contains the results?

Tables: All abbreviations should be explained in the captions. Tables are not clear

Results: In my opinion Authors should separate the effects of cardiac rehabilitation on pain perception from psychosocial factors.

The Authors did not indicate any limitations in this study.

Reviewer 3 Report

This is a paper which evaluates pain perception and tolerance in cardiac patients during rehabilitation. Hypothesis and aims are not clearly stated. Results are not supported by the Methods. English language and style also requires extensive editing. Abstract is also poorly written, not structured and it does not contain numbers about the results.